# Biallelic mutations in valyl-tRNA synthetase gene *VARS* are associated with a progressive neurodevelopmental epileptic encephalopathy

Jennifer Friedman[1,2,3,4], Desiree E. Smith[5,6], Mahmoud Y. Issa [7], Valentina Stanley[1,8], Rengang Wang[1,8], Marisa I. Mendes[5,6], Meredith S. Wright[4], Kristen Wigby[3,4], Amber Hildreth[3,4], John R. Crawford[1,2,3], Alanna E. Koehler [8], Shimul Chowdhury[4], Shareef Nahas[4], Liting Zhai[9], Zhiwen Xu[9,10], Wing-Sze Lo[9,10], Kiely N. James[1,8], Damir Musaev[1,8], Andrea Accogli[11,12], Kether Guerrero[11,13,14], Luan T. Tran[11,13,14], Tarek E.I. Omar[15], Tawfeg Ben-Omran[16], David Dimmock[4], Stephen F. Kingsmore[4], Gajja S. Salomons[5,6], Maha S. Zaki[7], Geneviève Bernard[11,13,14] & Joseph G. Gleeson[1,2,3,4,8]

Aminoacyl-tRNA synthetases (ARSs) function to transfer amino acids to cognate tRNA molecules, which are required for protein translation. To date, biallelic mutations in 31 ARS genes are known to cause recessive, early-onset severe multi-organ diseases. *VARS* encodes the only known valine cytoplasmic-localized aminoacyl-tRNA synthetase. Here, we report seven patients from five unrelated families with five different biallelic missense variants in *VARS*. Subjects present with a range of global developmental delay, epileptic encephalopathy and primary or progressive microcephaly. Longitudinal assessment demonstrates progressive cortical atrophy and white matter volume loss. Variants map to the VARS tRNA binding domain and adjacent to the anticodon domain, and disrupt highly conserved residues. Patient primary cells show intact VARS protein but reduced enzymatic activity, suggesting partial loss of function. The implication of *VARS* in pediatric neurodegeneration broadens the spectrum of human diseases due to mutations in tRNA synthetase genes.

[1] Department of Neurosciences, University of California San Diego, La Jolla, CA 92093, USA. [2] Division of Child Neurology, Rady Children's Hospital, San Diego, CA 92123, USA. [3] Department of Pediatrics, University of California San Diego, La Jolla, CA 92093, USA. [4] Rady Children's Institute for Genomic Medicine, Rady Children's Hospital, San Diego, CA 92123, USA. [5] Department of Clinical Chemistry, Metabolic Unit, Amsterdam UMC (University Medical Centers), Vrije Universiteit Amsterdam, 1081 HV Amsterdam, The Netherlands. [6] Gastroenterology & Metabolism Amsterdam Neuroscience, 1081 HV Amsterdam, The Netherlands. [7] Department of Clinical Genetics, National Research Centre, Cairo 12311, Egypt. [8] Department of Neurosciences, Howard Hughes Medical Institute, University of California, San Diego, La Jolla, CA 92093, USA. [9] IAS HKUST-Scripps R&D Laboratory, Institute for Advanced Study, Hong Kong University of Science and Technology, Clear Water Bay, Kowloon, Hong Kong, China. [10] Pangu Biopharma, Edinburgh Tower, The Landmark, 15 Queen's Road Central, Hong Kong, China. [11] Departments of Neurology and Neurosurgery, Pediatrics and Human Genetics, McGill University, Montreal H3A 0G4, Canada. [12] IRCCS Istituto Giannina Gaslini, Genova 16147, Italy. [13] Division of Medical Genetics, Montreal Children's Hospital, McGill University Health Center, Montreal H4A 3J1, Canada. [14] Child Health and Human Development Program, Research Institute of the McGill University Health Center, Montreal H4A 3J1, Canada. [15] Department of Pediatrics, Alexandria University, Alexandria 21526, Egypt. [16] Clinical and Metabolic Genetics, Department of Pediatrics, Hamad Medical Corporation, 3050 Doha, Qatar. These authors contributed equally: Jennifer Friedman, Desiree E. Smith. These authors jointly supervised this work: Geneviève Bernard, Joseph G. Gleeson. Correspondence and requests for materials should be addressed to J.G.G. (email: jogleeson@ucsd.edu)

The spectrum of genes responsible for pediatric neurodevelopmental and neurodegenerative conditions (PNDDC) is extraordinarily broad, often requiring a whole-exome or whole-genome sequencing approach rather than a targeted gene or panel approach in order to achieve a molecular diagnosis and to identify new causes of disease. This is in part due to the vast number of genetic forms of PNDDC, in addition to wide-ranging phenotypic and genetic heterogeneity among these conditions. Previous publications have addressed this issue by recruitment of large cohorts of patients with PNDDC, or through matchmaker exchanges to identify recurrently mutated genes[1–5].

Charged tRNAs are required by the ribosome to meet the cellular requirement for protein synthesis, and their availability regulates many aspects of cellular function[6]. Charging of tRNAs requires the function of aminoacyl-tRNA synthetases (ARSs), which are ubiquitously expressed and highly evolutionarily conserved. Each amino acid has one or more designated ARS enzymes to catalyze a bond with a cognate tRNA. With 37 ARS genes for the 20 typical amino acids, 17 encode a cytoplasmic-localized enzyme, 17 encode a mitochondrial-localized enzyme, and 3 charge tRNA in both cellular locations[7]. Individual ARS proteins that function predominantly in the mitochondria are designated with a '2' after the gene name, for instance the VARS paralogue functions in the cytoplasm and VARS2 paralogue functions in the mitochondria to charge valine tRNAs.

Mutations in 31 different ARS genes are currently implicated in recessive diseases, with mutations in most mitochondrial-localized ARS genes associated with mitochondrial encephalopathies, and cytoplasmic-localized ARS genes associated with a wider range of diseases but affecting primarily the nervous system. Analysis of mutant ARS enzyme activity typically show moderate to severe reduction, but with some activity remaining[8–10], consistent with the idea that these are non-redundant and essential enzymes.

Valine is an essential branched chain amino acid, utilized for protein synthesis both in the cytoplasm and mitochondria. VARS2 is nuclear-encoded and is mutated in individuals with recessive mitochondrial encephalomyopathy[11–13]. Missense mutations in VARS2 found in patients can lead to protein destabilization, which can reduce steady-state protein levels. Karaca et al. identified VARS as a candidate gene among 40 other genes profiled in 128 families with a range of neurologic presentations. They reported two pathogenic variants NM_006295.2:c.2653C>T p.(Leu885Phe) and NM_006295.2:c.3173G>A p.(Arg1058Gln) in two consanguineous families with epileptic encephalopathy; phenotypic spectrum and function were not studied[3].

Here we expand the findings of Karaca et al. by using exome or genome sequencing to identify and characterize five distinct bi-allelic VARS variants in seven additional epileptic encephalopathy patients. The VARS phenotype is characterized by a spectrum of global developmental delay, epileptic encephalopathy and primary or progressive microcephaly. Pathogenic variants disrupt highly conserved residues, and localize to the VARS tRNA binding domain and adjacent to the anticodon domain. In patients for whom tissue was available, fibroblasts show intact VARS protein but reduced enzymatic activity, suggesting partial loss of function.

## Results

### Identification of VARS biallelic mutations in five families.
Patients were referred for neurological or genetic assessment for PNDDC at the National Research Center in Cairo, Rady Children's Hospital in San Diego, and McGill University Health Center in Montreal. Seven patients from five families were included in this study. Whole-exome (WES) or whole-genome (WGS) sequencing in each family led to the identification of a single genetic variant that met criteria for causality using established protocols[14–16]. The researchers enrolling subjects from these three locations were matched through collaborative networks and the Matchmaker Exchange[1] and recognized a common core phenotype of neurodevelopmental disorder with microcephaly, seizures, and cortical atrophy (NDMSCA), which contributed to the OMIM entry #617802 linked to VARS. Families 2937, 3308, and 3439, all from Egypt, had documented first-cousin parental consanguinity. Family 3007, originating from Syria, had a history of remote parental consanguinity. Family GB31, from Montreal, Canada, was Caucasian and had no history of consanguinity (Fig. 1a).

WES was used for analysis of families 2937, 3308, 3439, and GB31, and focused on identification of homozygous or compound heterozygous, rare (low minor allele frequency in the gnomAD variome (http://gnomad.broadinstitute.org) and not identified in the GME variome (http://igm.ucsd.edu/gme/)) and potentially damaging variants prioritized using the American College of Medical Genetics organization, OMIM identity, gene function, protein expression, phenotypic assessment, and in silico prediction tools. WGS was used for analysis of family 3007 with analysis filtered to retain variants with allele frequencies of <0.5% in the Exome Variant Server, 1000 Genomes, and gnomAD databases and were prioritized using the phenotypes extracted from the medical record using Human Phenotype Ontology codes. VAAST and Phevor prioritization algorithms were used to rank variants (Supplementary Methods 1).

For all families we used standard clinical variant filtering (Supplementary Methods 1), identifying a group of potentially pathogenic variants (Supplmentary Data 1). For families 2937, 3308, and 3439, with documented consanguinity, we additionally filtered for homozygous variants, and further filtered for segregation within the family, predicted damaging by 'Mutation Taster', and Polyphen2 score >0.85; only the variant in VARS remained for each family after this approach. For family 3007, five variants passed filtering for further consideration, and all except homozygous variants in VARS were discarded due to poor phenotypic overlap, failure to find a second variant or deletion for an autosomal recessive condition, or variant inherited from an unaffected parent in an autosomal dominant condition. For this family we were unable to fully exclude impact of heteroyzgous SCO2 stop-gain on clinical phenotype. For family GB1, four variants met filtering criteria and all except VARS were excluded based upon failure to find a second variant or deletion for an autosomal recessive condition or occurrence of variant in a healthy population. For this family, VARS remained as the final candidate, and bi-allelic inheritance was confirmed following co-segregation analysis.

Families 2937, 3308, and 3439, derived from a cohort of 5000 families that underwent WES, and two families recruited from Cairo, shared the same NM_006295.2:c.3355C>T p.(Arg1119Cys) variant (Families 3308 and 3439), suggesting a founder mutation (Table 1, Fig. 1a). One additional family from Cairo (Family 2937) with a similar phenotype had a NM_006295.2:c.2840G>A p.(Arg947His) variant. Family 3007, identified in San Diego and of remotely consanguineous Syrian descent, was one of 200 families that were studied with WGS, and had a NM_006295.2:c.1981C>A p.(Pro661Thr) variant. Family GB31 from Montreal, Canada had compound heterozygous NM_006295.2:c.(2074G>C;1324C>T) p.(Ala692Pro;Arg442*) variants. Candidate variants were confirmed by Sanger sequencing, with segregation confirmed amongst genetically informative family members according to a strictly recessive inheritance mode.

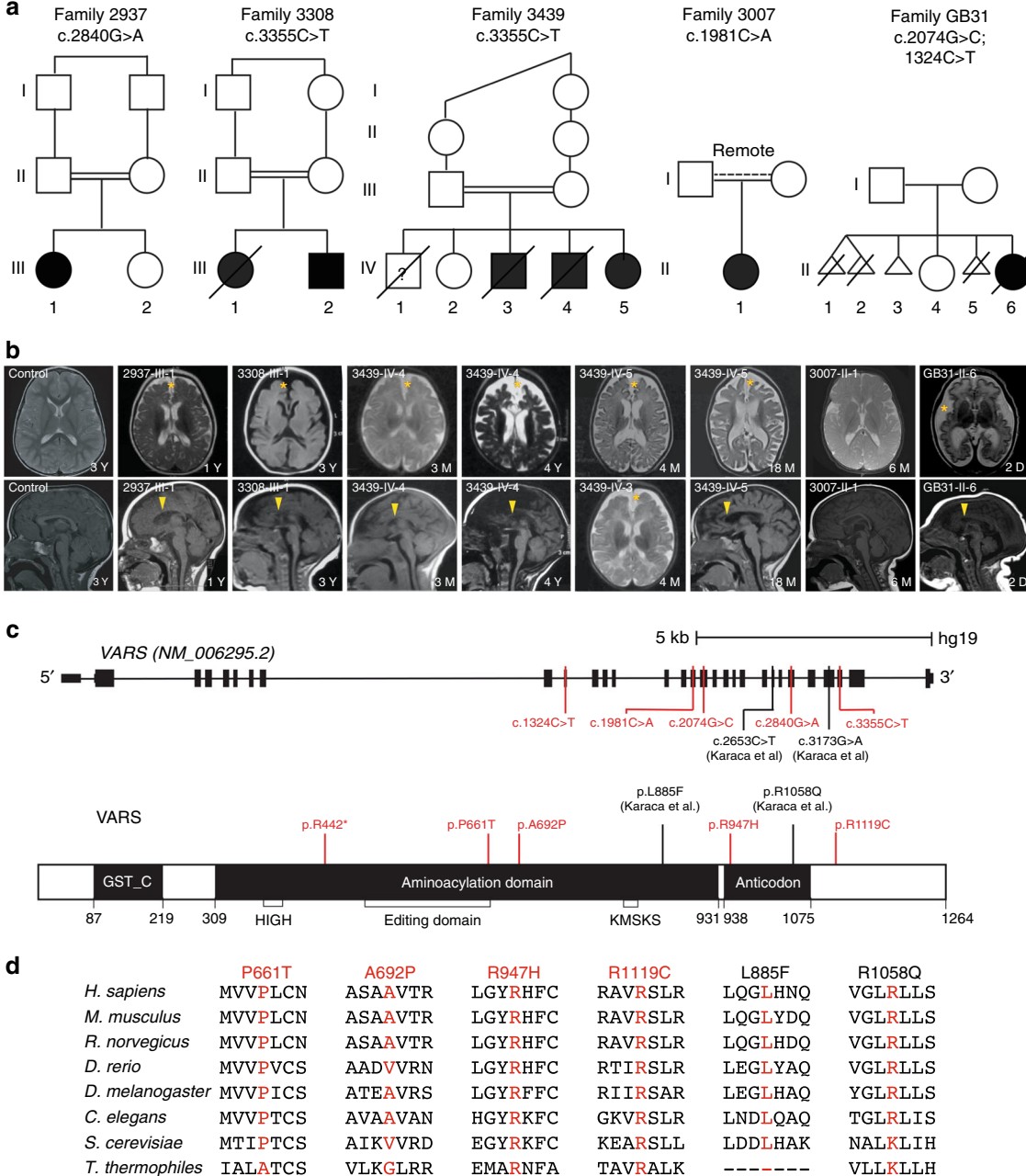

**Fig. 1** *VARS* mutations lead to a neurodevelopmental disorder with microcephaly, seizures, and cortical atrophy. **a** Pedigrees showing consanguinity in three families (double bars). Family 3007 showed remote consanguinity. Seven affected individuals were identified (black), and one individual was likely affected but DNA was not available for genotyping (3439-IV-1). Multiple miscarriages in family GB31 suggested fetal demise. **b** Available axial or midline sagittal brain MRIs demonstrated diffuse cortical atrophy (yellow star) and thinned corpus callosum (yellow arrow) compared to control. The brain MRI in 3007-II-1 was acquired at 6 months of age, and thus the patient was probably too young to observe these features. Two individuals from family 3439 had follow-up MRIs that demonstrated progressive cortical atrophy. D—days; M—months; Y—years. **c** Major domains of the 1264 amino acid protein, showing the GST, aminoacylation (including HIGH, Editing and KMSKS domains) and anticodon-binding domains. Disease-associated variants described in this study (red) and those from Karaca et al.[3] (black) are noted. Variants were in the aminoacylation or in or near the anticodon-binding domains, suggesting altered function. **d** Sequence conservation of mutated amino acids. The variants were either precisely conserved or maintain charge/polarity, with the exception of NM_006295.2:c.2653C>T p.(Leu885Phe), which is absent in *T. thermophiles*

## VARS mutations associate with a severe neurodevelopmental disorder.

Family 2937 had a single affected child from an uncomplicated delivery that developed generalized myoclonic and focal seizures at 3 months of age. Examination at 1 yr 7 months showed globally delayed milestones with head circumference of 40 cm (−4.6 SD), axial hypotonia and limb hypertonia, brisk reflexes and extrapyramidal movements. MRI performed at 1 yr showed cerebral cortical atrophy and mild central atrophy, with hypoplastic corpus callosum (Fig. 1b). Feeding difficulties persisted, but seizures had been well controlled on Valproate. Electroencephalogram showed right greater than left epileptiform discharges.

**Table 1 Case phenotype and genotype**

| | 2937-III-1 | 3308-III-1 | 3439-IV-3 | 3439-IV-4 | 3439-IV-5 | 3007-III-1 | GB31-II-6 |
|---|---|---|---|---|---|---|---|
| *Proband* | | | | | | | |
| Current age | 5 yrs F | Dec (3 yrs) | Dec (8 mos) | Dec (10 yrs) | 1.5 yrs | 10 mos | Dec (2.5 mos) |
| Gender | F | F | M | M | F | F | F |
| Ethnicity | Egyptian | Egyptian | Egyptian | Egyptian | Egyptian | Syrian | Caucasian |
| Consang | + | + | + | + | + | Distant | − |
| *Variant* | | | | | | | |
| Zygosity | Homozygous | Homozygous | N/A | Homozygous | Homozygous | Homozygous | Compound heterozygous |
| Genomic (hg19) | chr6:g.31748303C>T | chr6:g.31747247G>A | N/A | chr6:g.31747247G>A | chr6: g.31747247G>A | chr6:g.31750317G>T | chr6:g. (31750138C>G;31753046G>A) |
| cDNA | c.2840G>A | c.3355C>T | N/A | c.3355C>T | c.3355C>T | c.1981C>A | c.(2074G>C; 1324C>T) |
| Protein | p.(Arg947His) | p.(Arg1119Cys) | N/A | p.(Arg1119Cys) | p.(Arg1119Cys) | p.(Pro661Thr) | p.(Ala692Pro;Arg442*) |
| *Perinatal history* | | | | | | | |
| Gestation | Term | Term | Term | Term | Term | Term | Preterm (36 + 3 wks) |
| HC at birth (cm) | 33.5 (−0.9 SD) | 30 (−3.3 SD) | 31.5 (−2 SD) | 32.5 (−1.7 SD) | 32 (−1.25 SD) | 37 (+2.64 SD) | 30 (−2.06 SD) |
| Weight at birth (kg) | 2.53 (−1.7 SD) | 2.8 (−1 SD) | N/A | 3 (−1 SD) | 2.8 (−1.3 SD) | 3.55 (+0.67 SD) | 2.31 (−1.01 SD) |
| Length at birth (cm) | 46 (−1.6 SD) | 47 (−1 SD) | N/A | 48 (−0.6 SD) | 48 (−0.7 SD) | 53.3 (+2.25 SD) | 41 (−2.64 SD) |
| Complic | − | − | − | − | − | − | oligohyd; IUGR; seizures |
| *Psychomotor development* | | | | | | | |
| Gross motor skills | Delayed | Severe delay | Severe delay | Severe delay | Severe delay | Delayed | Severe delay |
| Fine motor skills | Delayed | Severe delay | Severe delay | Ssevere delay | Severe delay | Delayed | N/A |
| Language | Delayed | Severe delay | Delayed | Severe delay | Delayed | Delayed | N/A |
| Social | Delayed | Severe delay | Severe delay | Severe delay | Severe delay | Delayed | N/A |
| *Neurological examination* | | | | | | | |
| Age at last exam | 1 yr, 7 mos | 6 mos | 7 mos | 6 yrs | 1.5 yrs | 10 mos | 2 mos |
| HC (cm) | 40 (−4.6 SD) | 36 (−5.2 SD) | 38.2 (−4 SD) | 39 (−8.2 SD) | 37.5 (−6.8 SD) | 46.2 (+1.4 SD) | 30.5 (−6.5 SD) |
| Weight (kg) | 8 (−2.6 SD) | 5.5 (−2.1 SD) | 5.8 (−2.5 SD) | 7.7 (−5.8 SD) | 6.5 (−4.4 SD) | 9.5 (+0.85 SD) | 2.1 (−4.5 SD) |
| Length (cm) | 74 (−2.1 SD) | 63 (−0.8 SD) | 65 (−1.5 SD) | 97 (−3.6 SD) | 69 (−3.4 SD) | 69.5 (−0.97 SD) | 44 (−5.6 SD) |
| Tone | Axial hypotonia limb hypertonia | axial hypotonia limb hypertonia | axial hypotonia limb hypertonia | axial hypotonia, limb hypertonia | axial hypotonia limb hypertonia | axial hypotonia | axial and limb hypertonia |
| Reflexes | brisk | brisk | brisk | brisk | brisk | | brisk |
| Extrapyram | + | + | + | + | + | − | − |
| *Seizures* | | | | | | | |
| Onset | 3 mos | 3 wks | 2 wks | 1 wk | 40 d | 4 mos | 2 d |
| Type | Generalized, myoclonic, focal | Generalized, myoclonic, focal | Generalized, myoclonic, focal | Generalized, myoclonic, focal | Generalized, myoclonic, focal | Infantile spasms, tonic | Multifocal |
| Frequency | Rare | Weekly | Daily | Weekly | Daily | >10 per day | Daily |
| Control | Controlled | Refractory | Refractory | Refractory | Controlled | Controlled | Refractory |
| Treatments | Val | Val Lev | Val Lev Clon | Val Lev Clon | Lev Tia | Vig | Lev Phen Clon Lor |
| EEG | Right frontal epileptiform | Generalized discharges | Hypsarrhythmia | Generalized focus | Generalized discharges | Hypsarrhythmia | Diffuse multifocal epileptiform |
| *Brain MRI* | | | | | | | |
| Age of MRI | 1 yr | 3 yrs | 4 mos | 3 mos / 4 yrs | 4 mos / 18 mos | 6 mos | 2 d |
| Findings | Cortical atrophy, mild central atrophy, hypoplastic CC | Cortical atrophy, abnormal gyration, hypoplastic CC | Cortical atrophy, hypoplastic CC | Progressive central and cortical atrophy, hypoplastic CC | Progressive cortical atrophy, hypoplastic CC | Normal | Severe atrophy and simplified gyral pattern; abn myelination hypoplastic CC |
| *Systemic manifestations* | | | | | | | |
| Gastrointestinal | Feeding difficulties | Feeding difficulties, recurrent vomiting | Feeding difficulties, recurrent vomiting | Feeding difficulties, recurrent vomiting | Recurrent vomiting, AST slight elevation | Feeding difficulties, elevated AST/ALT, US: abn liver echo | Feeding difficulties, mild hepatomegaly |
| Other | | VSD | | | | | PFO, Skeletal survey abn*, Dysmorphic** |

+ present, − absent, *d* days, *mos* months, *wks* weeks, *yr(s)* years, *dec* deceased, *Consang* consanguinity, *HC* head circumference, *SD* standard deviations, *Complic* complications, *oligohyd* oligohydramnios, *IUGR* intrauterine growth restriction, *Extrapyram* extrapyramidal movements, *Val* Valproate, *Lev* Levetiracetam, *Clon* Clonazepam, *Tia* Tiagabine, *Vig* Vigabatrin, *Phen* Phenobarbital, *Lor* Lorazepam, *CC* corpus callosum, *AST* aspartate aminotransferase, *ALT* alanine aminotransferase, *US* ultrasound, *abn* abnormal, *echo* echogenicity, *VSD* ventriculoseptal defect, *PFO* patent foramen ovale
*11 ribs bilaterally, suspicion of congenital osteofibrous dysplasia of the right tibia
**Sloping forehead, prominent nasal bridge, apparent hypertelorism, upslanting and short palpebral fissure, large ears and a lumbar hemangioma

Family 3308 presented with a single affected child with seizures beginning at 3 weeks, characterized by generalized myoclonic and focal seizures on a weekly basis, refractory to treatment with Valproate and Levetiracetam. Electroencephalogram showed frequent generalized discharges. Assessment at 6 months showed severe global developmental delay, head circumference of 36 cm (−5.2 SD), axial hypotonia and limb hypertonia with brisk reflexes. There was a ventriculoseptal defect, which was clinically insignificant. The child died at 3 yr from chronic respiratory insufficiency resulting from neuromotor disability.

Family 3439 presented with four affected children, all showing global developmental delay, axial hypotonia, brisk reflexes, and extrapyramidal movements. The oldest (IV-1) died at 8 months from seizures and respiratory compromise, but otherwise followed a nearly identical clinical course. DNA was unavailable for family members IV-1 and IV-3, and thus molecular diagnosis could not be confirmed in these individuals. The electroencephalogram in 3439-IV-3 showed hypsarrhythmia, and seizures were refractory to triple therapy with Valproate, Levetiracetam,

and Clonazepam. At last exam, the youngest affected child was medically fragile but seizures were controlled with Levetiracetam and Tiagabine. All other affected individuals have now died.

Family 3007 presented with a single affected child at 4 months with global developmental delay and infantile spasms. Electroencephalogram met criteria for hypsarrhythmia. At 10 months the seizures were controlled on Vigabatrin monotherapy. MRI at 6 months was essentially unremarkable, and head circumference continued along normal percentiles, but global developmental delay was evident. This child displayed feeding difficulties and had persistent mild elevations of liver transaminases (aspartate aminotransferase and alanine aminotransferase in the low 100s, without impaired coagulation or gamma-glutamyl transferase levels). Liver ultrasound demonstrated mildly increased echogenicity, not otherwise specified, suggesting mild liver involvement, but follow-up had not demonstrated progression.

Family GB31 presented with a severely affected child soon after birth (36 weeks gestation). The mother had a history of recurrent pregnancy loss, including an elective termination at 18 weeks

(twins) due to meningomyelocele, a spontaneous termination at 8 weeks, and an elective termination at 14 weeks due to anencephaly. GB31-II-6 presented at birth with microcephaly, severe axial hypotonia and appendicular hypertonia, whole body clonus and seizures. Electroencephalogram demonstrated diffuse multifocal epileptiform abnormalities. Despite therapy with Levetiracetam, Phenobarbital, Clonazepam, and Lorazepam, seizures continued. The child also had facial dysmorphisms including a sloping forehead, prominent nasal bridge, hypertelorism, upslanting and short palpebral fissure, large ears, and small lumbar hemangiomas. Skeletal survey showed 11 ribs bilaterally and congenital osteofibrous dysplasia of the right proximal tibia. Brain MRI at birth showed severe cortical and cerebellar atrophy with simplified gyral pattern, delayed myelination, and hypoplastic corpus callosum. By 2 months of age, head circumference was 30.5 cm ($-6.5$ SD), there was axial and limb hypertonia, severe feeding difficulties, a clinically significant patent foramen ovale, and hepatomegaly. Respiratory insufficiency ensued and the child passed away at 2.5 months of age. Neuropathological assessment in GB31-II-6 showed brain weight of 169 g ($-6$ SD) with simplified gyral pattern, marked reduction in cerebral white matter volume, absence of cavitation or necrotic lesions, but thin corpus callosum. The hemispheric white matter was intensely gliotic with minimal myelination but preserved axonal density, with a few sudanophilic histiocytes, consistent with a myelination disorder. Hippocampus, basal ganglia, thalamus, brainstem, and cerebellum were histologically unremarkable.

**VARS deficiency associates with progressive cortical atrophy.** We noted that the head circumference of most affected children tended to slow in growth by 1 yr of age (Table 1), which predicted a loss in cerebral growth and/or atrophy. To confirm this suspicion, we repeated brain MRI in older children to assess for acquired cortical atrophy. Individual 3439-IV-4 had initial brain MRI at 3 months and then repeated at 4 yr (Fig. 1b). Side-by-side comparison demonstrated progressive loss of cerebral cortical volume, with collapse of subcortical white matter volume, evident on both axial and midline sagittal images. Similarly, individual 3439-IV-5 had an initial brain MRI at 4 months and then repeated at 18 months, demonstrating loss of subcortical white matter volume and increased lateral ventricle size. The MRI findings together with brain neuropathology from GB31-II-6 demonstrate that pathogenic variants in VARS also show particular predilection for the cerebral white matter.

**Mutations in VARS alter evolutionarily conserved amino acids.** The VARS mutations occurred throughout the open reading frame, predicting alteration in evolutionarily conserved amino acids. VARS encodes a 1264 amino acid protein consisting of an N-terminal glutathione-S-transferase and aminoacylation enzymatic domains and a C-terminal anticodon-binding domain, used for recognition of valine tRNAs (Fig. 1c). The amino acids that we identified as mutated in our cohort, as well as one of the pathogenic variants identified by Karaca et al.[3] were conserved in all mammals and were further conserved to within the same amino acid type in all species profiled. Furthermore, all of our patients had variants that altered amino acid charge or group (Fig. 1d). The NM_006295.2:c.2653C>T p.(Leu885Phe) Karaca et al. mutation, however, had no counterpart in *T. thermophiles*.

**Variants predicted to affect protein structure or function.** The class Ia ARS enzymes, which include VARS, utilize an alpha-helix bundle domain to recognize the key residues in the anticodon domain of the cognate tRNA, essential for correct charging[17]. The structure of the VARS orthologue has been determined in the

tRNA-bound state for *T. thermophiles*[18], where tRNA residues A35-C36 (the major identity elements), are base-stacked upon each other, and fit into a pocket on the alpha-helix bundle domain of VARS. Hydrogen bonds are formed between VARS and A35-C36 of tRNA^Val in a base-specific manner. The discrimination of valine from the structurally similar isoleucine and threonine in the amino acid binding domain occurs via 'double sieve' mechanisms[19], whereby VARS excludes these incorrect bases using distinct mechanisms.

To predict the impact of the mutations on protein structure, we utilized a crystal structure of the *T. thermophiles* VARS-tRNA^Val complex because human and *T. thermophilies* protein sequences share high sequence similarity (~90% conservation)[18,19], and it is the only species where structure is available. We constructed a model of human VARS-tRNA^Val complex by homology modeling and examined the location relative the alpha-helical and beta-sheets of the protein (Supplementary Fig. 1), as well as how each mutation is predicted to impact local structure and tRNA binding (Fig. 2a). Residues Pro661, Arg947, and Arg1119 are in proximity to the tRNA molecule (<6 Å). The p.(Pro661Thr) mutant exhibits no obvious change in distance from tRNA or in contacts with adjacent amino acids, but occurs within a conserved beta-sheet. Both Arg947 and Arg1119 are in alpha helices and in close proximity to the C-37 and C-39 of the tRNA^Val, respectively. These may create hydrogen-bonding interactions that would be abolished by the mutations, possibly leading to reduced tRNA binding (Figs 2b, d, f). Arg947 in particular has been previously hypothesized to be important in tRNA binding[18].

The other two residues, Ala692 and Arg1058, have a distance of >15 Å from the tRNA, but again occur in alpha helices. The NM_006295.2:c.2074G>C p.(Ala692Pro) introduces the helix breaker proline and is predicted to result in a loss of a hydrogen bond with Ala689 (Fig. 2c), which may cause a conformational change, while the NM_006295.2:c.3173G>A p.(Arg1058Gln) mutation effect is less certain (Fig. 2e). Thus, all patient mutations that we identified occur within highly ordered regions of VARS, and some were additionally predicted to have an impact on tRNA binding or protein structure, possibly affecting the VARS function.

**Patient mutations in VARS reduce aminoacylation activity.** To evaluate the impact of variants on protein stability and function, we collected skin punches then generated primary fibroblast cultures from an affected member of four of the families, as well as parental healthy controls (obligate carriers) in three families, and compared with results from cultures from unrelated healthy controls. We observed that cells from affected individuals displayed typical growth characteristics under basal conditions. For some of the recessive ARS genes mutated in human disease, there are reduced levels of protein at steady state, including for the mitochondrial-localized VARS2[13,20–23]. However, western blot analysis demonstrated VARS protein levels in patient cells that were indistinguishable from controls (Fig. 3a), suggesting that these mutations do not destabilize VARS or alter steady-state levels in detectable ways.

VARS transferase activity was assessed by incubating cytoplasmic extracts with yeast total tRNA, ATP and [$^{15}$N]-valine. Valine bound by tRNA was subsequently quantified by LC-MS/MS. LARS (leucine) and RARS (arginine) activities were assessed simultaneously as controls. Extracts from affected individuals (2937-III-1, 3308-III-1, 3439-IV-4, GB31-II-6) showed that VARS transferase activity was reduced by about fivefold (i.e., to 20% of levels observed in controls, Fig. 3b) which proved to be statistically significant ($P < 0.0001$ for both comparisons, one tailed Student's $t$-test). This reduced but not abolished activity is

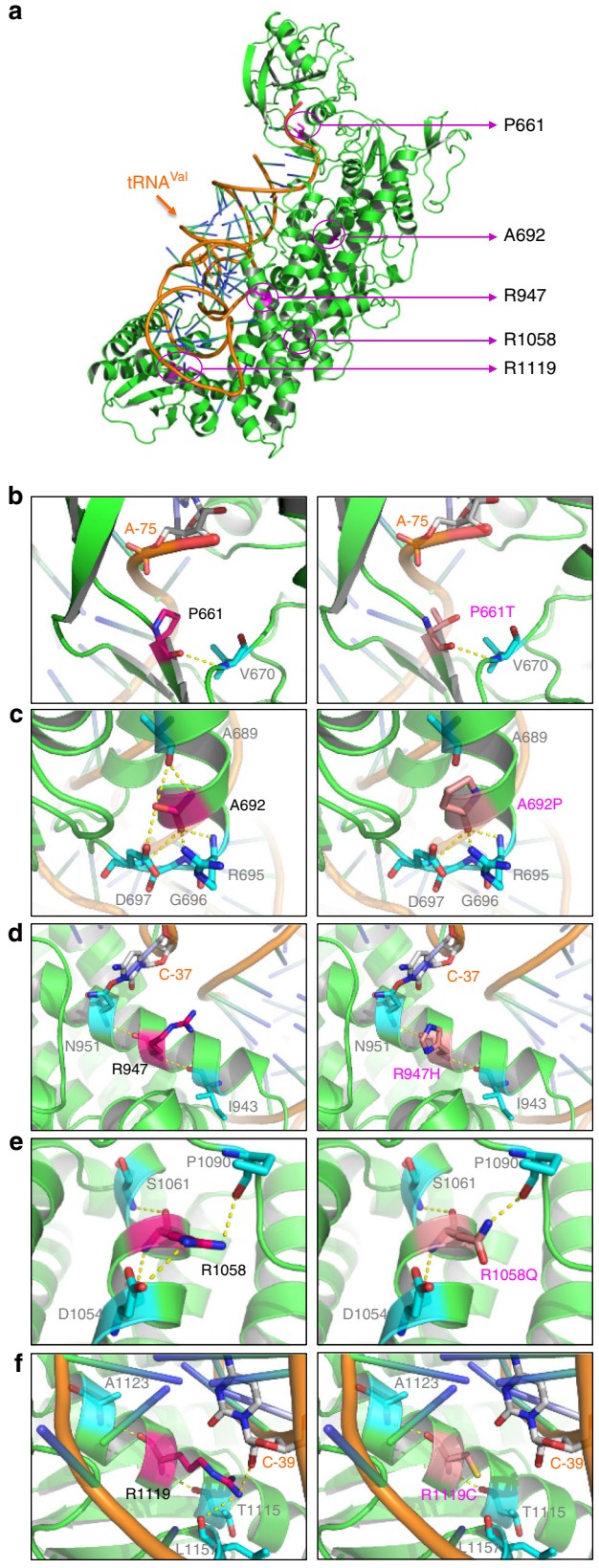

**a**

**Fig. 2** Predicted mutational impact on VARS structure and tRNA binding. **a** Locations of the mutational sites on the structural model of human VARS-tRNA^Val complex (VARS green, tRNA^Val orange, mutational sites magenta). Residues Pro661, Arg947, and Arg1119 are closer to the tRNA molecule (<6 Å), while Ala692 and Arg1058 have a longer distance of >15 Å from tRNA. **b**–**f** Pair-wise comparisons between the wild-type (left) and mutant (right) residues for predicted changes in local contacts with tRNA or other amino acids. Amino acids are indicated by letter and number (e.g., Pro661), tRNA bases are indicated by letter and dash and number (e.g., A-75). Hydrogen bonds are presented as yellow dashed lines. **b** Residue Pro661 forms a hydrogen bond with Val670 and has a distance of ~4.5 Å from A-75 of the tRNA. **c** Residue Ala692 forms hydrogen bonds with adjacent residues Ala689, Arg695, Gln696, and Asp697. Mutation to proline predicts abolished contacts with Ala689 and disrupted helical structure. **d** Residue Arg947 is close to C-37 and predicts hydrogen-bonding interaction that is abolished by the Arg947H mutation. This mutation does not predict altered hydrogen-bonding with Ile943 and Asn951. **e** Residue Arg1058 forms hydrogen bonds with Asp1054, Ser1061 and Pro1090. The NM_006295.2: c.3173G>A p.(Arg1058Gln) mutation predicts elimination one of the two possible bonds with Asp1054. **f** The side chain of Arg1119 forms a hydrogen bond with C-39, predicted to be abolished by the mutation

60–80% of unrelated control levels (Fig. 3c), which is consistent with their heterozygous state. Controls all showed activities within 10% of each other. Activities of LARS and RARS in patients showed some variability in activity but were all within 75–110% of control activities. These experiments point to a specific reduction in VARS activity in patient cells, which likely accounts for their clinical presentations.

## Discussion

The process of charging tRNAs with amino acids is an evolutionarily conserved process in all metazoa, critical for protein translation. In mammals there are 37 ARS proteins linking the 20 primary amino acids with the 497 human tRNAs. In this study, we identified seven patients from five families with biallelic mutations in *VARS*. There were five pathogenic variants in total, with three of these present in a homozygous state, while two were seen in a compound heterozygous state, which associated with the most clinically severe case. All missense variants were absent or reported only rarely in public references databases (EXAC, gnomAD), predicted pathogenic by in silico analysis, disrupted highly conserved amino acid residues in critical regions of the protein, and led to severely reduced transferase activity in the tested patient cells.

The five distinct *VARS* pathogenic variants reported here and two previously identified[3] were distributed throughout the VARS protein. All variants were encompassed in the full-length transcript subserving tRNA synthetase activity. Five were located in the aminoacylation domain, and two were in or adjacent to the anticodon-binding domain. Interestingly, irrespective of where the mutation occurred, we found reduced aminoacylation activity, suggesting that transferase activity reflects both the ability to bind tRNA as well as transfer the charged valine to the tRNA.

Recent attention has focused on non-translational function of ARS proteins, evidenced by the addition of newly evolved domains that are neither necessary nor sufficient for ARS activity[24]. For the most part, the function of these additional protein domains, some of which are mammalian-specific, remain unknown. The mammalian-specific VARS N-terminal 218 residue GST domain is not essential for ARS activity[25], but rather binds the elongation factor EF-1α to facilitate association and

consistent with reduced activity seen in other recessive ARS diseases. Next we determined if healthy parents showed reduced activity, given that they are carriers (i.e., heterozygous) for the mutation. We found that parents had activities that ranged in the

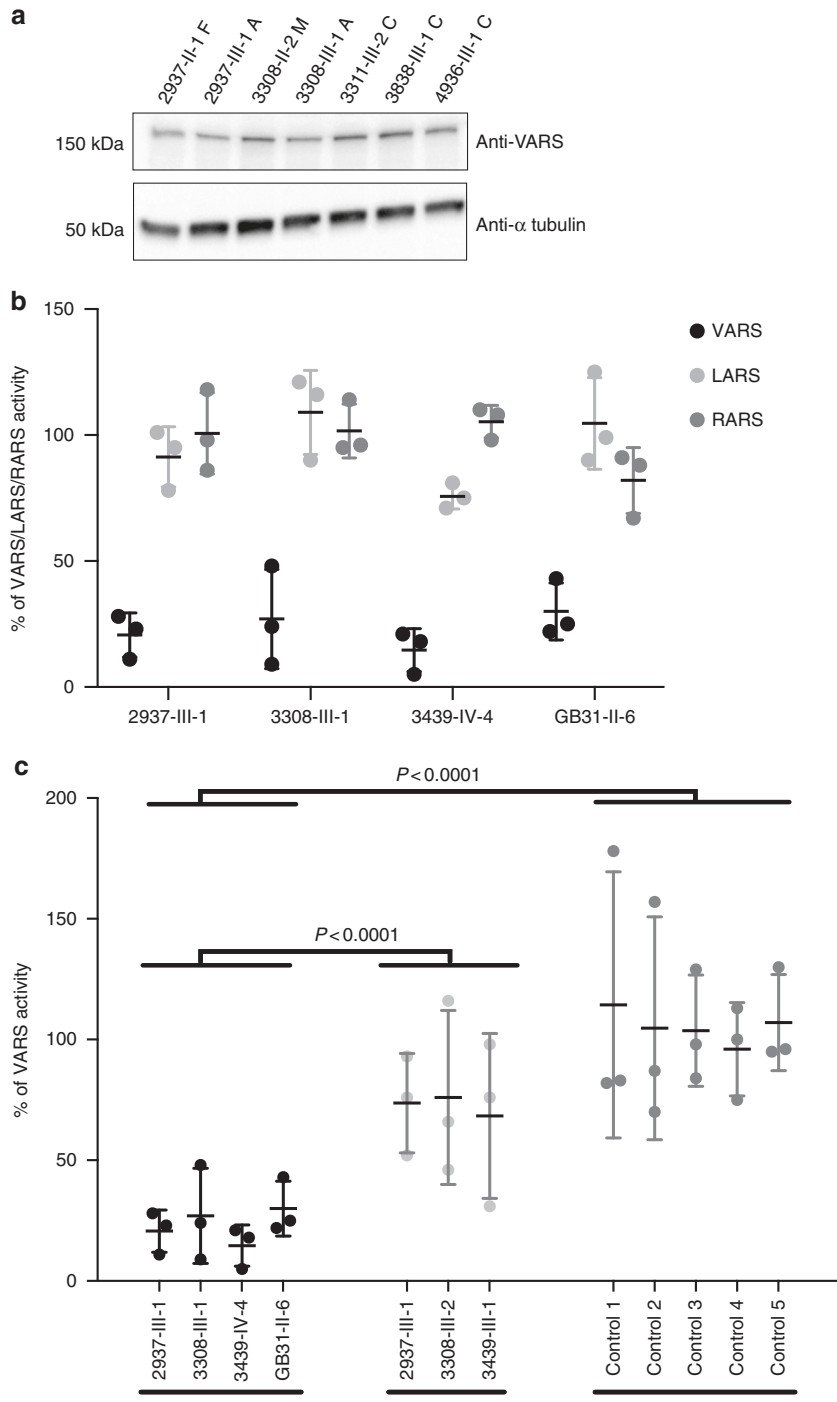

**Fig. 3** *VARS* patient cells show normal VARS protein level, but reduced enzymatic ARS activity. **a** Western blot of patient or control cells showing retained VARS protein level in affected individuals (2937-III-1A and 3308-III-1A), carrier parents (2937-II-1F and 3308-II-2M), and controls (3311-III-2C, 3838-III-1C, and 4936-III-1C). **b** VARS, LARS (leucine ARS) and RARS (arginine ARS) aminoacylation activity. VARS activity was specifically reduced in patient cells, normalized to control cells at 100% (from $n = 3$ replicates, error bars = S.E.M.). **c** VARS activity in patients compared with carrier parents and controls. Black: patients, light gray: parents of affecteds (obligate carriers of the *VARS* mutation), dark gray: healthy controls (from $n = 3$ replicates, error bars = S.E.M.). Differences between patients and controls or carriers were calculated based upon the average of the group ($P < 0.0001$ for both comparisons, One tailed Student's $t$-test). Full gel in Supplementary Fig. 2

efficient transfer of charged tRNAs[26]. In addition, there are at least six alternatively spliced *VARS* transcripts, which are catalytic nulls, but display specific biological activities across a spectrum of cell-based assays, particularly 'neutrophil oxidative burst'[27]. Whether any of the mutations we identified additionally interfere with non-catalytic functions of VARS will require further study.

Mutations in both mitochondrial and cytoplasmic-localized *ARS*s cause a spectrum of recessive disorders, many of which include neurological phenotypes. Pathogenic variants in cytoplasmic *ARS*s have been associated with microcephaly, seizures, developmental delay, intellectual disability, and sensorineural hearing loss[28]. *RARS*, *DARS*, and *EPRS* bi-allelic mutations cause

hypomyelination[29–31], *SARS* mutations are implicated in intellectual disability[32], and *HARS* has been associated with Usher syndrome[33]. Dominant (mono-allelic) mutations in multiple ARSs genes (*GARS*, *YARS*, *AARS*, and *HARS*) have been associated with the progressive axonal neuropathy Charcot Marie Tooth type 2 Disease[33–38], presumably acting in a dominant-negative capacity. In addition to neurologic phenotypes, other systemic manifestations are associated with recessive cytoplasmic *ARSs* mutations (*LARS*, *MARS*), including liver dysfunction and interstitial lung disease[39,40]. The mechanism driving tissue-specific phenotypes associated with certain *ARS* genes remains largely unexplored.

The non-syndromic clinical presentation of these affected individuals prevented us from identifying them as the same genetic condition prior to WES or WGS, but in retrospect several clinical aspects are shared. The clinical presentation in the first weeks to months of life with severe epilepsy despite aggressive anticonvulsant treatment, and association with hypsarrhythmia on electroencephalogram can be a clue. Another clue is acquired microcephaly, in the range of minus 5 SDs, observed in all but one subjects after 6 months of age. The brain MRI appearance of cortical atrophy with reduced white matter volume without edema, and without involvement of deep brain structures or brainstem volume loss can differentiate *VARS* from leukodystrophies, mitochondrial, and pontocerebellar disorders. Unfortunately, none of our subjects have yet survived past childhood, and most succumb from medical complications in the first few years of life, suggesting a lethal condition.

The clinical features of neurodevelopmental delay with microcephaly, seizures, and cortical atrophy (MIM#617802) linked with *VARS* mutations can be compared with other defined conditions, as well as the function of the genes implicated in these disorders. Recessive microcephaly, seizures, and developmental delay (MCSZ MIM#613402) due to mutations in the DNA-repair enzyme encoded by *PNKP* has onset in infancy. Recessive microcephaly (postnatal progressive) with seizures and brain atrophy (MIM#613668) is due to mutations in *MED17*, encoding a subunit of the transcription pre-initiation mediator complex. The *VARS* phenotype is also similar to *AIMP1* related disorders (MIM# 603605) which present with microcephaly, cortical cerebral atrophy and early-onset seizures but can be differentiated from *VARS* based upon presence of hypomyelinating leukodystrophy, similar to that of X-linked Pelizaeus–Merzbacher disease[41,42]. While there are over 55 unique genetic causes for early infantile epileptic encephalopathy (EIEE), many involving neuronal ion channels, we think that the *VARS*-related syndrome can be distinguished clinically from EIEE.

The etiology of varied features and severity in our cohort are uncertain. These may include environmental factors, although the contribution of other rare variants identified in these patients cannot be excluded at this point. Multiple miscarriages occurring in family GB31 that included two different neural tube defects suggests that the phenotypic spectrum may be even broader, and, at the extreme end, incompatible with life. Unfortunately, the deceased fetuses were not available for genotyping, so it remains unproven if they shared the same genetic mutations. Notably this family displayed the only protein-termination variant in our cohort, which not surprisingly may have a more severe impact on protein function.

There is emerging evidence that genes encoding factors involved in RNA metabolism including tRNA synthetase, tRNA splicing complex, and RNA helicases[43–46], play essential roles in human brain development. Further study including in vitro and in vivo modeling systems will be necessary to determine the mechanism by which pathologic *VARS* variants disrupt tRNA synthetase activity, whether alternatively splice transcripts have a role in neural development or function, how these abnormalities disrupt normal brain development and how these *VARS* functions may be related to the function of other genes implicated in neural development.

## Methods

**Patient ascertainment**. All experiments involving human participants or data were conducted in compliance with all relevant ethical regulations. Informed consent was obtained from all participants and their families. Approval for human subjects research was obtained from National Research Center in Cairo and University of California, San Diego Institutional Review Boards, the McGill University Health Center Reasearch Institute Research Ethics Board, and Rady Children's Hospital Research Compliance. All affected individuals were clinically evaluated by a pediatric neurologist and geneticist. Subjects underwent complete dysmorphological, neurological, and general examinations. Brain MRI was obtained in all living subjects, and longitudinal assessment was performed up until the present time to determine disease course. Venous blood or saliva for DNA isolation was collected from all genetically informative living members of each family at the time of ascertainment. In four families, dermal skin punch was obtained after consent from the index individual and genetically informative relatives, and cultured in 10% fetal bovine serum and DMEM to obtain low-passage primary fibroblast cultures.

**Whole exome or genome sequencing and variant identification**. Details regarding sequencing, filtering and prioritization protocols for each family are outlined in Supplementary Note 1 and Supplementary Data 1. Exome or Genome sequencing was performed on affected member(s) from each family using established methods[16,47,48]. All variants were prioritized by allele frequency, conservation, and predicted effect on protein function, and were tested by Sanger sequencing for segregation with disease within the entire family.

**Construction of human VARS-tRNA$^{Val}$ structural model**. The human VARS structure model was generated by the protein structure homology-modeling server Phyre V2.0[49] based on the *T. thermophiles* VARS structure (PDB: 1GAX). The structure of *T. thermophiles* tRNA$^{Val}$ molecule (PDB: 1GAX) was docked into the human VARS model using the PathDock server[50]. All the structural figures were prepared with Pymol (http://www.pymol.org).

**Western blot**. Western blots were performed with rabbit polyclonal anti-VARS antibody (Abclonal A4182; 1:1000) and mouse monoclonal anti-a-tubulin (Sigma T6074; 0.2 µg/ml) in 5% non-fat milk in PBST. Full blot in Supplementary Fig. 2.

**Combined VARS, LARS, and RARS activity assay**. Skin punches were collected at a single timepoint from each patient studied here, then expanded and used for enzymatic analysis. Steady-state aminoacylation assays were performed in technical triplicate in fibroblast lysates [cytosolic fraction, no mitochondrial proteins were detected using western blot (Supplementary Fig. 3)] at 37 °C for 10 min in reaction buffer (50 mmol/L Tris buffer pH 7.5, 12 mmol/L MgCl$_2$, 25 mmol/L KCl, 1 mg/mL bovine serum albumin, 0.5 mmol/L spermine, 1 mmol/L ATP, 0.2 mmol/L yeast total tRNA, 1 mmol/L dithiotreitol, 0.3 mmol/L [$^{15}$N]-valine, [$^{13}$C$_2$]-leucine and [$^{15}$N$_2$]-arginine). Reaction was terminated using trichloroacetic acid. After sample washing with trichloroacetic acid, ammonia was added to release the labeled amino acids from the tRNAs. [$^{13}$C]-valine, [D$_3$]-leucine, and [$^{13}$C$_6$]-arginine were added as internal standards. Labeled amino acids were quantified by LC-MS/MS. Intra-assay variation was <15%. Student *t*-test with unequal variances were performed using SPSS version 22 to test significant differences between patients and controls/carriers.

## Data availability

*VARS* variant data that support the findings of this study have been deposited in ClinVar with accession codes [https://www.ncbi.nlm.nih.gov/clinvar/variation/522588/#clinical-assertions] SCV000715349.1 (family 3007), SCV000734850 (c.2074G>C), and SCV000734851 (c.1324C>T) (family GB31). The whole-exome sequencing data from families 2937, 3308, and 3439 in this study have been deposited in the database of Genotypes and Phenotypes (dbGaP) under accession phs000288.v1.p1. Additional data that support the findings of this study are available from the corresponding author upon reasonable request.

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

## Acknowledgements

We thank the patients and their families for participation. This study was supported by grants from the NIH (1R01NS098004 to J.G.G.) and QNRF (NPRP 6-1463-3-351 to J.G. G. and T.B.-O., the Research Grants Council of Hong Kong (16100015 to Z.X.), the Rady Children's Institute for Genomics Medicine for sequencing and bioinformatic support, the Yale University Center for Mendelian Genomics (Mark Gerstein, Murat Gunel, Richard P. Lifton, Shrikant M. Mane, 5UM1HG006504), the Broad Institute (Daniel G. MacArthur and Heidi L Rehm, 5UM1HG008900), the Canadian Institutes of Health Research (201610PJT-377869), the Fondation du Grand Défi Pierre Lavoie, Fondation les Amis d'Eliott, Fondation Lueur d'Espoir pour Ayden, and the Réseau de Médecine Génétique Appliquée of the Fonds de Recherche du Québec en Santé. We also wish to acknowledge the McGill University and Génome Québec Innovation Centre. This research was enabled in part by support provided by Compute Canada (www.computecanada.ca). G.B. has received a Research Scholar Junior 1 award from the Fonds de Recherche du Québec en Santé (FRQS) (2012-2016) and the New Investigator Award from the Canadian Institutes of Health Research (2017-2022). We thank Rady Children's Institute for Genomic Medicine Investigators: Yan Ding, Lisa Salz, Kelly Watkins, and Sergey Batalov. We also thank Drs. Xiang-Lei Yang, Leslie A. Nangle, and Litao Sun for helpful comments and advice, Steffen Albrecht for providing neuropathological assessment, and Peter de Jonghe for sharing unpublished data. The authors would also like to thank the Exome Aggregation Consortium and the groups that provided exome variant data for comparison. A full list of contributing groups can be found at http://exac.broadinstitute.org/about.

## Author contributions

J.F., M.Y.I., V.S., J.C., K.N.J., K.W., A.H., T.O., M.S.Z., G.B., and J.G.G. recruited patients, gathered clinical data and described the clinical phenotype. D.E.S., M.I.M., S.C., R.W., M. W., J.F. A.K., S.N., L.Z., Z.X., W.-S.L., A.A., K.G., L.T.T., and G.S.S. performed experiments and analysis, D.M. and M.W. identified mutations in human *VARS* gene, J.F., G.B.,

and J.G.G. wrote the manuscript. T.B.-O., G.S.S., G.B., J.F., and J.G.G. supervised the project. All authors critically reviewed the paper.

## Additional information

**Competing interests:** D.D. is a consultant for Biomarin, Ichorian, and Complete Genomics. J.F. holds shares in Illumina and Spouse is Founder and Principal of Friedman Bioventure, which holds a variety of publicly traded and private biotechnology interests. D.D. is on the Scientific Advisory Board for Audentes Therapeutics. The remaining authors declare no competing interests.

