## [Peer Review File · Nature Communications]

Reviewer #1 (Remarks to the Author):

With interest I read the manuscript by Friedman et al reporting on bi-allelic VARS mutations causing a progressive neurodevelopmental epileptic encephalopathy. The authors provide a concise and well-written manuscript containing both clinical and molecular genetic data of patients with mutations in VARS.

I have only some minor comments that the authors could consider to improve their manuscript.

- the mutation nomenclature is not according the HGVS
- Figure 1a (or c) would benefit from showing which mutations were identified in which patients
- From the introduction paragraph it is not clear that the two families by Karaca are not included in the families presented in the manuscript, but that these were only used for 'comparison' reasons both for the mutations identified, as well as for the clinical phenotype observed. (see also line 130, where it, in my opinion, seems to be that there has been extensive contact with the Karaca team)
- Figure 1b: for non-clinicians, it would be good to indicate the abnormalities observed in the MRIs by for instance arrows
- It could be considered to combine Figures 2 and 3 into one.

Reviewer #2 (Remarks to the Author):

Friedman and colleagues provide strong genetic evidence to support a link between homozygosity of missense mutations in VARS and severe neurological microcephalic developmental disorders. Although this is a very focused and convincing set of data showing that the patients carrying two modified copies of the gene have dramatic loss of enzymatic activity of VARS, the data does not show that this reduction to 20% of activity is the cause of their global developmental delay, epileptic

encephalopathy and primary or progressive microcephaly. Overall, the discovery of the link between VARS and the complex developmental syndrome is of great interest for Nature Comms readership. However, some functional link between low enzymatic activity and neurological developmental abnormalities need to be presented to fully make the case for causality.

- In Figure 1b, the MRI panels need to highlight the aspects affected compared to normal human MRI at these ages. Nature comms scientific readership is broad and the images as presented now are not very meaningful to most readers. A similar set of axial and sagittal views of normal brains (at least for 1 and 4 years) should be shown for comparison.
- The enzymatic activity of the patients cell extracts should show control levels in Figure 2. I would prefer to see Suppl. Figure 2 data on heterozygous incorporated in the actual Figure 2.
- Some basic evaluation of cell cycle and neurogenic potential of the iPSCs from the patients would be necessary to show that human embryos can't develop normally with 20% activity of their VARS proteins.

Reviewer #3 (Remarks to the Author):

Friedman et al. provide some evidence, suggesting that mutations in VARS the only known valine cytoplasmic-localized aminoacyl-tRNA synthetase may be responsible for a patient phenotype characterized by neurodevelopmental deficits and epilepsy. They expand the known range of mutations in this gene that are connected with this kind of phenotype. The first association of VARS with NEURODEVELOPMENTAL DISORDER WITH MICROCEPHALY, SEIZURES, AND CORTICAL ATROPHY was presented by Karaca et al. in 2015 (<https://omim.org/entry/192150>).

The genetic findings presented in this paper are based on WGS or WES. These technologies frequently lead to the discovery of more than one variant that co-segregate with the disease and may have pathogenic potential. This is why filtering and prioritization of variants are extremely important and the strategy used should be addressed in detail. In this respect the manuscript in its present form is in my view not specific enough. I would like to see more information about the sequencing results and the bioinformatic analysis, including the most prominent variants/genes that were filtered out in the different individuals. This is interesting because the phenotype varies somewhat (see also below) and there might be putative additional factors among the filtered out variants. This should also be addressed in the discussion. Also, since different effect prediction

algorithms often result in different predictions concerning the effects of specific sequence variants, I suggest that the authors use a variety of effect prediction programs (e.g. SIFT, PolyPhen, VEP, SNAP2, MutationTaster) for interpretation of their results, presented perhaps in a table for easy comparison by the reader. This is all the more important because, even though the patient phenotype is more or less well overlapping among the investigated individuals, there are also differences. For example in patient 3007, who does not show microcephaly but also differences concerning the type and frequency of seizures, and it would be interesting if there are also differences at the genetic level i.e. severity of the impairment or other features of the mutation that could explain this. Particularly, since for this patient there is no experimental evidence for a loss of VARS function presented either.

Experimental findings concerning VARS activity should be presented with proper information on the statistics performed and the level of significance should be indicated. It is not clear in which way the intra-assay variation is taken into account. Also, it is not clear whether technical or biological replicates were investigated.

The discussion of the spectrum of ARS-related disorders (p.8) is not quite up to date since recent publications on the topic, as e.g. by Musante et al. (Mutations of the aminoacyl-tRNA-synthetases SARS and WARS2 are implicated in the etiology of autosomal recessive intellectual disability. *Hum Mutat.* 2017 Jun;38(6):621-636. doi: 10.1002/humu.23205. Epub 2017 Mar 23.) are not mentioned.

Re: Rebuttal for Re: NCOMMS-18-00330

REVIEWER #1

-With interest I read the manuscript by Friedman et al reporting on bi-allelic VARS mutations causing a progressive neurodevelopmental epileptic encephalopathy. The authors provide a concise and well-written manuscript containing both clinical and molecular genetic data of patients with mutations in VARS.

- Response. We thank Reviewer #1 for these positive comments.

- I have only some minor comments that the authors could consider to improve their manuscript. - the mutation nomenclature is not according the HGVS

- Response. Corrected

Figure 1a (or c) would benefit from showing which mutations were identified in which patients

- Response.
 - Mutations have been added to figure 1 to improve clarity.
 - Figure 1b – MRI abnormalities have been highlighted and compared to normal MRI (as requested by Reviewer #2)

-From the introduction paragraph it is not clear that the two families by Karaca are not included in the families presented in the manuscript, but that these were only used for 'comparison' reasons both for the mutations identified, as well as for the clinical phenotype observed. (see also line 130, where it, in my opinion, seems to be that there has been extensive contact with the Karaca team).

- Response: Families described in Karaca et al. were presented for comparison purposes only. The text has been edited to clarify that data related to patients described in Karaca et al. was not included in this study.

- It could be considered to combine Figures 2 and 3 into one.

- Response: We thank the reviewer for suggestion to streamline our figures by combining Figures 2 and 3. We have chosen not to do this as Reviewer #2 has requested instead enlarging Figure 3 by incorporating Supplemental Data Figure 2 into Figure 3.

REVIEWER #2

-Friedman and colleagues provide strong genetic evidence to support a link between homozygosity of missense mutations in VARS and severe neurological microcephalic developmental disorders. Although this is a very focused and convincing set of data showing that the patients carrying two modified copies of the gene have dramatic loss of enzymatic activity of VARS, the data does not show that this reduction to 20% of activity is the cause of their global developmental delay, epileptic encephalopathy and primary or progressive microcephaly. Overall, the discovery of the link between VARS and the complex developmental syndrome is of great interest for Nature Comms readership. However, some functional link between low enzymatic activity and neurological developmental abnormalities need to be presented to fully make the case for causality.

- Response: We thank Reviewer #2 for these positive comments and address individual points below.

- In Figure 1b, the MRI panels need to highlight the aspects affected compared to normal human MRI at these ages. Nature commss scientific readership is broad and the images as presented now are not very meaningful to most readers. A similar set of axial and sagittal views of normal brains (at least for 1 and 4 years) should be shown for comparison.

- Response: Figure 1b has been modified to highlight the MRI abnormalities compared with normal child brain as a control. We have included images from a single 3 year old child as normal imaging are similar for the ages of affected patients presented.

- The enzymatic activity of the patients cell extracts should show control levels in Figure 2. I would prefer to see Suppl. Figure 2 data on heterozygous incorporated in the actual Figure 2.

- Response. Done

- Some basic evaluation of cell cycle and neurogenic potential of the iPSCs from the patients would be necessary to show that human embryos can't develop normally with 20% activity of their VARS proteins.

- Response. We did not generate patients' iPSCs in this study and are therefore unable to study their cell cycle and neurogenic potential. We agree with Reviewer #2 that these experiments would be interesting in the future but are beyond the scope of this paper. In this study, our efforts were focused on demonstrating the pathogenicity of the patient missense mutations, by identifying enzymatic defects in patient cells, as demonstrated in the text and figures.

REVIEWER #3

-Friedman et al. provide some evidence, suggesting that mutations in VARS the only known valine cytoplasmic-localized aminoacyl-tRNA synthetase may be responsible for a patient phenotype characterized by neurodevelopmental deficits and epilepsy. They expand the known range of mutations in this gene that are connected with this kind of phenotype. The first association of VARS with NEURODEVELOPMENTAL DISORDER WITH MICROCEPHALY, SEIZURES, AND CORTICAL ATROPHY was presented by Karaca et al. in 2015 (<https://omim.org/entry/192150>).

The genetic findings presented in this paper are based on WGS or WES. These technologies frequently lead to the discovery of more than one variant that co-segregate with the disease and may have pathogenic potential. This is why filtering and prioritization of variants are extremely important and the strategy used should be addressed in detail. In this respect the manuscript in its present form is in my view not specific enough. I would like to see more information about the sequencing results and the bioinformatic analysis, including the most prominent variants/genes that were filtered out in the different individuals. This is interesting because the phenotype varies somewhat (see also below) and there might be putative additional factors among the filtered out variants. This should also be addressed in the discussion. Also, since different effect prediction algorithms often result in different predictions concerning the effects of specific sequence variants, I suggest that the authors use a variety of effect

prediction programs (e.g. SIFT, PolyPhen, VEP, SNAP2, MutationTaster) for interpretation of their results, presented perhaps in a table for easy comparison by the reader. This is all the more important because, even though the patient phenotype is more or less well overlapping among the investigated individuals, there are also differences. For example in patient 3007, who does not show microcephaly but also differences concerning the type and frequency of seizures, and it would be interesting if there are also differences at the genetic level i.e. severity of the impairment or other features of the mutation that could explain this. Particularly, since for this patient there is no experimental evidence for a loss of VARS function presented either.

- Response. We now present specific information about how variants were prioritized and filtered. Filtering and prioritization of variants was performed in three independent labs. The issue of variant prioritization is simplified in our consanguineous families, because there were very few homozygous rare potentially damaging variants to consider that fell into regions of homozygosity, which was the case for 3 of the families. Nevertheless, in every family, the VARS gene emerged as the top candidate based upon heuristic variant prioritization, which we now describe in the manuscript. Details regarding the bioinformatics analysis including description of the most prominent genes/variants that were considered for each individual are summarized in the supplemental data section. A table summarizing in silico predictions for each variant using several prediction programs (SIFT, PolyPhen, VEP, SNAP2, MutationTaster) has been added as a Supplemental Table.
- We agree that etiology of variable phenotypic features and severity in our cohort is of great interest. There may be genetic and/or environmental factors influencing phenotypic severity in our patients. Interestingly, as seen in several neurodegenerative diseases, the age of onset of the disease is inversely proportional to the rapidity of neurodegeneration. Though we are unable to provide any definitive explanation for the variable clinical features in our cohort, the influence of genetic modifiers is of great interest. Unfortunately, our cohort is too small to draw conclusions regarding possible interactions. We have nevertheless, added a statement regarding possible genetic or environmental modifiers to the discussion.

-Experimental findings concerning VARS activity should be presented with proper information on the statistics performed and the level of significance should be indicated. It is not clear in which way the intra-assay variation is taken into account. Also, it is not clear whether technical or biological replicates were investigated.

- Response. Greater details of the experimental conditions are now provided in the methods section. Skin punches were collected at a single time from each patient studied here, then expanded and used for enzymatic analysis. We have performed all transferase activity assays according to standard clinical procedures, in triplicate technical replicates, but without biological replicates. The intra-assay variation of an analytical method displays which part of the variation between samples can be explained by the method itself. Hence any difference between control and patient that is larger than the intra-assay variation (15%) must be due to an altered VARS activity of the patient sample. Since technical triplicates were measured for this study, this analytical variation is already taken into account.

-The discussion of the spectrum of ARS-related disorders (p.8) is not quite up to date since recent publications on the topic, as e.g. by Musante et al. (Mutations of the aminoacyl-tRNA-synthetases SARS and WARS2 are implicated in the etiology of autosomal recessive intellectual disability. Hum Mutat. 2017 Jun;38(6):621-636. doi: 10.1002/humu.23205. Epub 2017 Mar 23.) are not mentioned.

- Response. We agree and have incorporated these recent results into our manuscript.

Reviewer #1 (Remarks to the Author):

I would like to start by thanking the authors for the revised version of their manuscript. Taking into account the comments of all reviewers, I believe the manuscript has been improved substantially.

I have two small remarks remaining. Firstly, the HGVS nomenclature uses '(' instead of '['. Please adjust. Secondly, in figure 3c, the authors use p-values to indicate their statistical significant values obtained. Please use $p < 0.0001$ (or provide the actual obtained scientific value) instead $p = 0.000$ for scientific accuracy.

Reviewer #2 (Remarks to the Author):

I am happy with the response to my comments.

Reviewer #3 (Remarks to the Author):

I appreciate that in response to my earlier comment, the authors have now presented the range of variants found in the respective pedigrees.

However, I am still not convinced by the presented evidence that (functional) VARS-deficiency is the sole cause of the observed phenotype in each of the presented families.

While it is of course true that VARS shows up in all families, in each there are variants in other genes that are considered damaging by some or even a majority of prediction tools. In one case a nonsense mutation in a different gene was found but completely disregarded. The change affects SCO2 in 3007-III-1, the patient with the most divergent phenotype. The mutation has the potential to be functionally severe and in my opinion cannot be disregarded solely based on the absence of a reported second mutation for an autosomal recessive condition or poor phenotypic overlap with previous findings. Particularly the latter appears a little ironic since the phenotypic overlap of this patient with the others is also far from perfect and no results concerning VARS activity are shown for this patient. Experiments addressing the molecular effects of this change and their combination with the impact of the VARS alteration would tremendously enhance the novelty and originality of this

study. As it stands, however, neither the clinical nor the molecular basis for including this patient are very convincing.

In Family 3439 a duplication was found in GPR88 and one splice region variant in MLKL. Not surprisingly both do not have a score from the prediction tools used, however, these changes might well have pathogenic potential. Both manuscript and supplement, however, fail to address these specific variants and why they were discarded without further analysis. I would expect at least an in silico prediction of the effect of the insertion on the affected protein structure, an experimental test of the potential splice mutation to rule out its functional impact and proper discussion of their exclusion.

Specific points:

The article merely reports the coincidence of VARS variants and a specific phenotype without functional data that are sufficient to warrant a statement such as “VARS deficiency results in progressive cortical atrophy”. This needs to be toned down e.g. “VARS deficiency is associated with progressive cortical atrophy”.

NCOMMS-18-00330B

Response to reviewer comments

Reviewer #1 (Remarks to the Author):

I would like to start by thanking the authors for the revised version of their manuscript. Taking into account the comments of all reviewers, I believe the manuscript has been improved substantially.

Response. We thank Reviewer #1 for these positive comments.

I have two small remarks remaining. Firstly, the HGVS nomenclature uses '(' instead of '['. Please adjust. Secondly, in figure 3c, the authors use p-values to indicate their statistical significant values obtained. Please use $p < 0.0001$ (or provide the actual obtained scientific value) instead $p = 0.000$ for scientific accuracy.

Response. We have corrected to HGVS '(' designation and $p < 0.0001$.

Reviewer #2 (Remarks to the Author):

I am happy with the response to my comments.

Response. We thank Reviewer #2 for this positive comment.

Reviewer #3 (Remarks to the Author):

I appreciate that in response to my earlier comment, the authors have now presented the range of variants found in the respective pedigrees. However, I am still not convinced by the presented evidence that (functional) VARS-deficiency is the sole cause of the observed phenotype in each of the presented families. While it is of course true that VARS shows up in all families, in each there are variants in other genes that are considered damaging by some or even a majority of prediction tools. In one case a nonsense mutation in a different gene was found but completely disregarded. The change affects SCO2 in 3007-III-1, the patient with the most divergent phenotype. The mutation has the potential to be functionally severe and in my opinion cannot be disregarded solely based on the absence of a reported second mutation for an autosomal recessive condition or poor phenotypic overlap with previous findings. Particularly the latter appears a little ironic since the phenotypic overlap of this patient with the others is also far from perfect and no results concerning VARS activity are shown for this patient. Experiments addressing the molecular effects of this change and their combination with the impact of the VARS alteration would tremendously enhance the novelty and originality of this study. As it stands, however, neither the clinical nor the molecular basis for including this patient were very convincing. In Family 3439 a duplication was found in GPR88 and one splice region variant in MLKL. Not surprisingly both do not have a score from the prediction tools used, however, these changes might well have pathogenic potential. Both manuscript and supplement, however, fail to address these specific variants and why they were discarded without further analysis. I would expect at least an in silico prediction of the effect of the insertion on the affected protein structure, an experimental test of the potential splice mutation to rule out its functional impact and proper discussion of their exclusion.

Response. We thank Reviewer #3 for providing this useful feedback. We agree that there are several other variants that appear from exome sequencing in the affected members of the families, which have the potential to influence phenotype. In the revised manuscript we consider this potential and indicate that 'a contribution of the additional rare variants cannot be fully excluded at this point.' In consultation with the Editor around this suggestion, we now discuss these data within the body of the manuscript.

We apologize for not including additional information about predicted effect of the individual variants on protein or protein function. The ‘stop-gain’ variant in SCO2 in family 3007 was inherited from the asymptomatic father and was present as heterozygous in the affected child, without evidence for a mutation on the child’s other chromosome. SCO2 encodes cytochrome c oxidase assembly protein. Biallelic loss of function mutations associates with cardioencephalomyopathy (OMIM 604377) whereas dominant loss associates with myopia (OMIM 608908), and thus do not phenotypically overlap with our subject. While we cannot completely exclude a contribution of this variant to the VARS phenotype, there is no evidence of cardiac disease in the child, documented by a normal echocardiogram. We additionally reevaluated the WGS to verify that there was no second mutation or INDEL on the other chromosome. As we were unable to fully exclude a contribution of this variation to the phenotype, we have added a sentence acknowledging this to the text. The variant in GPR88 in family 2937 leads to insertion of two alanine residues at a position 241 in the protein, where there are already other benign alanine insertions described in the gnomAD database including this same 2 alanine insertion in 2 presumed healthy individuals, as well as 1 alanine insertion in 9 individuals and 3 alanine insertions in 1 individual. Given the polymorphic nature of this part of the genome/protein, the predication software programs that deal with INDELS rank this polymorphism as non-damaging. Further we performed modeling of this mutation using the program PROVEAN (<http://provean.jcvi.org/>) and found that this variant has a PROVEAN score of -1.109 and is considered ‘neutral’. The variant in the MLKL occurs at position -7 before exon 3 and is predicted to have a minimal effect on splicing using the Human Splicing Finder 3.1 program (<http://umd.be/HSF3/>). The HSP score was 95.64 for wild-type and 95.22 for variant (out of 100), suggesting the variant has little effect on splicing.

Specific points:

The article merely reports the coincidence of VARS variants and a specific phenotype without functional data that are sufficient to warrant a statement such as “VARS deficiency results in progressive cortical atrophy”. This needs to be toned down e.g. “VARS deficiency is associated with progressive cortical atrophy”.

Response. We have now used the term ‘associated with’ rather than ‘results in’.

Reviewer #1 (Remarks to the Author):

I am happy with the responses to my comments. I however did notice that in line 286 of the manuscript $p=0.000$ has not been changed into $p<0.0001$.

Of note, in lines 262 (no space) and line 286 (full stop) typos are present.

September 8, 2018

We thank R1 for the positive comments and have addressed these in the manuscript as noted below.

REVIEWERS' COMMENTS:

Reviewer #1 (Remarks to the Author):

I am happy with the responses to my comments. I however did notice that in line 286 of the manuscript $p=0.000$ has not been changes into $p<0.0001$.

This has been corrected.

Of note, in lines 262 (no space) and line 286 (full stop) typos are present.

These typographical errors have been corrected.